# Association between Daily Urinary Sodium Excretion, Ratio of Extracellular Water-to-Total Body Water Ratio, and Kidney Outcome in Patients with Chronic Kidney Disease

**DOI:** 10.3390/nu13020650

**Published:** 2021-02-17

**Authors:** Kaori Kohatsu, Sayaka Shimizu, Yugo Shibagaki, Tsutomu Sakurada

**Affiliations:** 1Division of Nephrology and Hypertension, Department of Internal Medicine, St. Marianna University School of Medicine, Kawasaki, Kanagawa 216-8511, Japan; kaori.kohatsu112@gmail.com (K.K.); eugo@wc4.so-net.ne.jp (Y.S.); 2Department of Healthcare Epidemiology, Kyoto University, Kyoto 606-8507, Japan; shimizu.sayaka.5e@kyoto-u.ac.jp; 3Institute for Health Outcomes and Process Evaluation Research (iHope International), Kyoto 604-8006, Japan

**Keywords:** chronic kidney disease, daily salt intake, fluid overload, renal outcome

## Abstract

Whether dietary salt intake affects chronic kidney disease (CKD) progression remains unclear. We conducted a retrospective cohort study to analyze the effects of both daily salt intake (DSI) and volume status on renal outcomes in 197 CKD patients. DSI was estimated by 24-h urinary sodium excretion and volume status was assessed by the ratio of extracellular water (ECW) to total body water (TBW) measured by bioelectrical impedance analysis (BIA). We divided patients into two groups according to DSI (6 g/day) or median ECW/TBW (0.475) and compared renal outcomes of each group. Furthermore, we classified and analyzed four groups according to both DSI and ECW/TBW. The higher DSI group showed a 1.69-fold (95% confidence interval (CI) 1.12–2.57, *p* = 0.01) excess risk of outcome occurrence compared to the lower group. Among the four groups, compared with Group 1 (low DSI and low ECW/TBW), Group 3 (high DSI and low ECW/TBW) showed a 1.84-fold (95% CI 1.03–3.30, *p* = 0.04) excess risk of outcome occurrence; however, Group 2 (low DSI and high ECW/TBW) showed no significant difference. High salt intake appears to be associated with poor renal outcome independent of blood pressure (BP), proteinuria, and volume status.

## 1. Introduction

High intake of dietary salt is reportedly associated with various adverse health outcomes, such as stomach cancer [1], osteoporosis [2], and kidney stones [3]. However, more crucial issues are undoubtedly new onset or worsening of hypertension and the occurrence of cardiovascular events. High salt intake is a known risk factor for not only cardiovascular disease, but also renal function loss, and salt restriction plays a key role in dietary management for chronic kidney disease (CKD) patients [4]. Indeed, strict salt restriction has been associated with improvements in hypertension and urinary protein reduction [5,6].

Several studies have reported that high salt intake as calculated by measuring 24-h urinary sodium excretion or as estimated from spot urine excretion was associated with hypertension, increased urinary protein, and occurrence of cardiovascular disease (CVD) [7,8,9]. However, the effects of salt intake on CKD progression remain unclear. Although some studies have indicated a positive association between urinary sodium excretion and CKD progression [10,11], others have shown no such association [12,13,14]. In addition, few studies have evaluated the long-term effects of salt restriction on renal outcomes.

Furthermore, a high-salt diet induces sodium retention in CKD patients, leading to fluid overload that in turn contributes to hypertension. Moreover, fluid overload itself has been suggested to represent an independent prognostic factor for cardiovascular mortality or all-cause mortality, even after adjusting for blood pressure (BP) control or proteinuria [15,16,17]. Fluid overload is also reportedly associated with poor renal outcomes [18,19,20]. In those studies, however, how salt intake and fluid overload interacted in their association with renal outcomes was not evaluated sufficiently.

We therefore aimed to analyze the effects of both daily salt intake (DSI) as estimated from 24-h urinary sodium excretion and volume status as assessed by the ratio of extracellular water (ECW) to total body water (TBW) measured using bioelectrical impedance analysis (BIA) on renal outcomes in patients with stage 3–5 CKD (CKD staging was based on KDIGO 2012 clinical practice guideline for the evaluation and management of chronic kidney disease).

## 2. Patients and Methods

### 2.1. Study Design and Setting

The present study used a single-center, retrospective cohort design. The study protocol was approved by the ethics committee of St. Marianna University School of Medicine (approval no. 4942). The need for informed consent was waived because of the retrospective nature of the study. The study adhered to the principles of the Declaration of Helsinki (as revised in Fortaleza, Brazil, October 2013). In addition, study information was published on the internet, to provide patients with the opportunity to use the official departmental website to opt out of the study if they did not want their data used for research purposes.

### 2.2. Study Population

A total of 464 patients with stage 3–5 CKD who were hospitalized at St. Marianna University School of Medicine Hospital for education regarding CKD from January 2011 to April 2019 were included. Patients lacking variables needed for multivariate analysis, from whom a sufficient 24-h urine specimen could not be collected (total urine volume <400 mL/day, as the definition of oliguria), or for whom eGFR was not followed up after discharge were excluded. All of the required data were available from 204 of the 464 patients. Among these 204 patients, 3 patients who met the definition of oliguria and 4 patients whose eGFR could not be followed after discharge were excluded. This resulted in a total of 194 participants enrolled in this analysis.

### 2.3. Measurements

#### 2.3.1. Patient Characteristics

We obtained information for patients on admission from the medical records, including age, sex, body mass index (BMI), etiology of CKD, comorbidities (diabetes mellitus (DM) or CVD such as ischemic heart disease, cerebrovascular disease, or peripheral arterial disease), use of renin–angiotensin system (RAS) inhibitors or diuretics, and mean systolic blood pressure (SBP) as measured by ambulatory blood pressure monitoring (ABPM). Values for 24-h ABPM were obtained using an automatic ABPM device (TM-2431; A&D, Tokyo, Japan) during the day. We also obtained laboratory findings, including albumin, hemoglobin, eGFR, urine protein, and DSI estimated from 24-h urinary sodium excretion. The eGFR was calculated from the serum creatinine level, age, and sex using the formula recommended by the Japanese Society of Nephrology [21]. All 24-h urine specimens were collected from the day of admission to the following morning.

#### 2.3.2. Exposures

(a) Division according to DSI

DSI was estimated from 24-h urinary sodium excretion [22,23]. We used the following formula:

DSI (g/day) = 24-h urinary sodium excretion (mEq/L) × daily urine volume (L)/17

Patients were divided into two groups according to DSI with a cut-off level of 6 g/day as recommended in an evidenced-based clinical practice guideline for CKD [24].

(b) Division according to ECW/TBW

We measured TBW, ECW, and ECW/TBW by BIA. Patients were then divided into two groups according to median ECW/TBW.

The present study used a BioScan 920-II multifrequency bioelectrical impedance analyzer (Maltron Bioscan, Rayleigh, UK). The eight tactile electrodes were attached to the dorsum of the wrists and third metacarpi of both hands, and the anterior surfaces of the ankles and third metacarpi of both feet with the patient supine on a flat, nonconductive bed. The Bioscan analyzer allows multi-frequency measurement (5, 50, 100, and 200 kHz) with a low-amplitude current (700 μA). The data obtained included body fluid composition, separated into water-free mass consisting of proteins, fat, and minerals, TBW, intracellular water (ICW), and ECW. These measurements were performed by experienced laboratory technicians blinded to the background of enrolled patients.

(c) Division according to both DSI and ECW/TBW

Further, we classified four groups according to both DSI and ECW/TBW: Group 1, low DSI and low ECW/TBW; Group 2, low DSI and high ECW/TBW; Group 3, high DSI and low ECW/TBW; and Group 4, high DSI and high ECW/TBW.

#### 2.3.3. Outcomes

The primary outcome was defined as a ≥30% decline in eGFR from baseline (on admission) or occurrence of end-stage renal disease (ESRD, taken as initiation of renal replacement therapy (hemodialysis, peritoneal dialysis, or kidney transplantation)) or death. Survival time was calculated from enrollment (admission date) to occurrence of the event. Subjects lost to follow-up because of withdrawal or hospital transfer were censored as of the time of last visit, and those who had not shown any events as of 30 April 2020 were censored at that date.

#### 2.3.4. Statistical Analysis

Measured values are expressed as median (interquartile range (IQR)) or mean (SD), as appropriate. Categorical variables are described as frequency (*n*) and ratio (%). Correlations between DSI and both ECW/TBW and SBP were determined by Pearson’s correlation coefficients. Differences between the four groups divided according to both DSI and ECW/TBW were compared by analysis of variance for continuous normally distributed variables and using the Kruskal–Wallis test for continuous asymmetrically distributed variables. Categorical variables with expected frequencies below 10 were assessed using Fisher’s test, and all others were assessed by chi-squared analysis. Survival curves were drawn using the Kaplan–Meier method and the log-rank test was used for group comparisons. Proportional hazard assumption was confirmed by testing based on the Schoenfeld residual. Comparison of outcomes among groups shown in Section 2.3.2. was assessed using the hazard ratio (HR) calculated by Cox proportional hazards analysis. Multiple covariables were adjusted for age, sex, eGFR, hemoglobin, albumin, log urinary protein (UP), SBP, presence or absence of DM, and CVD. All statistical analyses were performed using Stata/MP version 16.1 software (StataCorp, College Station, TX, USA). Values of *p* < 0.05 were considered statistically significant.

## 3. Results

### 3.1. Baseline Characteristics

Demographic and clinical characteristics of a total of 194 patients are summarized in Table 1. Mean age was 70.5 (SD 12.1) years, and 75.6% were male. The vast majority of patients (94.4%) had hypertension, and 46.7% had DM. The most frequent cause of CKD was diabetic nephropathy (31.5%), followed by nephrosclerosis (27.9%). RAS inhibitors were used in 67.5% of patients, and diuretics in 31.5%. DSI estimated from 24-h urinary sodium excretion was 5.88 g (IQR 4.35–8.24 g), and mean ECW/TBW was 0.48 (SD 0.04). The median ECW/TBW of 0.475 was used to classify patients into two groups. These distributions are described in the histograms in Figure 1.

On the basis of DSI estimated from 24-h urinary sodium excretion and ECW/TBW, 62 patients were categorized as Group 1, 42 patients as Group 2, 37 patients as Group 3, and 56 patients as Group 4. Patients in Group 1 were significantly older and more frequently female and showed significantly lower BMI and UP than the other three groups (Table 1). Furthermore, etiologies of CKD in Group 1 were more frequently nephrosclerosis and chronic glomerulonephritis and less frequently diabetic nephropathy. In addition, a smaller number of people had hypertension in Group 1. Meanwhile, patients in Group 4 were significantly younger and more frequently male and had higher BMI and frequency of DM. Moreover, they showed significantly higher UP and DSI than the other groups. Irrespective of DSI, patients with ECW/TBW above the median showed significantly higher BMI, UP, prevalence of diabetes mellitus nephropathy, lower albumin, and more frequent use of RAS inhibitors and diuretics compared to those with ECW/TBW below the median.

### 3.2. Patient Outcomes

During follow-up (median, 1.4 years; IQR 0.7–2.4 years), 107 patients (54.3%) showed an outcome, namely, a ≥30% decline in eGFR in 49.7%, induction of ESRD in 3.0%, and death in 1.5%. The incidence rate of clinical outcomes was 29.8 per 100 person-years (Table 2).

### 3.3. Correlation of DSI and Each of ECW/TBW and SBP

DSI showed very weak correlations with mean SBP in ABPM (r = 0.24, *p* < 0.01) and ECW/TBW (r = 0.21, *p* < 0.01) (Figure 2).

### 3.4. Comparisons of Outcomes among Two Groups According to ECW/TBW or DSI

Higher DSI (>6 g/day) was significantly associated with outcomes compared with lower DSI (≤6 g/day) (HR 1.69, 95% CI 1.12–2.57; *p* = 0.01). Conversely, no significant association with clinical outcomes was seen between ECW/TBW groups.

### 3.5. Comparison of Clinical Background and Outcomes among Four Groups According to Both ECW/TBW and DSI

Survival curves for the four groups are shown in Figure 3. With log-rank testing, the *p*-value was 0.22. Cox proportional hazards analysis to compare outcomes among the four groups showed Group 3 (high DSI and low ECW/TBW) had a 1.84-fold (95% CI 1.03–3.30-fold; *p* = 0.04) excess risk of outcome occurrence compared with Group 1 (low DSI and low ECW/TBW). However, Groups 2 and 4 showed no significant differences compared with Group 1 (Figure 4).

## 4. Discussion

The present study investigated the effects of both DSI and volume status on renal outcome in patients with stage 3–5 CKD. Among two groups divided by DSI, higher DSI (>6 g/day) was significantly associated with renal outcome compared to lower DSI (≤6 g/day) (HR 1.69, 95% CI 1.12–2.57; *p* = 0.01). On the other hand, no significant association was identified between groups categorized by ECW/TBW. We then classified four groups based on both DSI and ECW/TBW. Among these groups, compared with Group 1 (low DSI and low ECW/TBW), Group 3 (high DSI and low ECW/TBW) showed a 1.84-fold (95% CI 1.03–3.30; *p* = 0.04) excess risk of outcome occurrence. Interestingly, Group 2 (low DSI and high ECW/TBW) displayed no significant difference compared with Group 1.

Several studies have demonstrated positive correlations between a high-salt diet and CKD progression [10,25,26]. In contrast, some studies have suggested no such association [12,13,14]. However, those studies showed limitations with specific subgroups such as non-diabetic patients [12], or type 1 DM [13], and advanced CKD patients [14], which could not sufficiently cover the general CKD population. In addition, some studies used spot urine samples to evaluate salt intake, which might have resulted in inaccurate estimations. He et al. [10] recently reported the association of 24-h urinary sodium excretion with CKD progression and all-cause mortality among 3757 patients with CKD in the Chronic Renal Insufficiency Cohort Study. In that study, the highest quartile of urinary sodium excretion (≥194.6 mmol/24 h) showed a 1.54-fold (95% CI 1.23–1.92-fold) excess risk of CKD progression compared with the lowest quartile (≤116.8 mmol/24 h). However, that association disappeared after adjusting for proteinuria, which suggested that proteinuria might represent an important mechanism underlying CKD progression associated with high salt intake. Proteinuria is known to be a significant risk factor for poor renal outcome, and previous studies have suggested that high salt intake enhances angiotensin-converting enzyme activity in renal tissues, which could reduce the effect of RAS blockers and lead to increased UP and subsequent poor renal outcomes [25]. On the other hand, Kang et al. [26] reported 24-h urinary sodium excretion was associated with CKD progression independent of BP or proteinuria, supporting the present results. The mechanisms by which high salt intake leads to kidney damage are gradually revealed. The direct effects of high salt intake on kidney damage have been investigated in various basic research studies. Experimental studies have shown that high salt intake induces intrarenal production of angiotensin II [27], stimulates the synthesis of pro-inflammatory cytokines [28], and increases oxidative stress [29] and inflammation, which might contribute to arterial stiffness and/or endothelial dysfunction. Recent studies have also reported that Rac1, a member of the Rho family GTPases activated by high salt intake, activates mineralocorticoid receptor without aldosterone, in turn inducing salt-sensitive hypertension, proteinuria, and glomerulosclerosis [28,29].

The present study suggested that high DSI as indicated by increased urinary sodium excretion could represent an independent risk factor for CKD progression aside from SBP and proteinuria, consistent with the results of previous basic clinical research [26,27,28,29,30,31].

In recent reports, fluid overload has been suggested to impact renal outcomes in CKD patients not receiving dialysis [18,19,20]. Those studies used various markers of volume status, such as ECW/TBW measured by BIA and level of overhydration (OH) calculated from the difference between measured ECW and normal expected ECW predicted using physiological models under euvolemic conditions [32]. Hung et al. also reported that fluid overload as assessed by OH/ECW was associated with renal outcome independent of BP [19]. The mechanism by which fluid overload induces progression of kidney disease has been revealed to include decreased renal blood flow due to increasing renal efferent pressure [33], arterial stiffness, endothelial activation, and inflammation [19,34]. Hung et al. [19] suggested that patients or animals with volume overload had significantly higher levels of proinflammatory cytokines such as interleukin 6 and tumor necrosis factor α compared to those without fluid overload. Other studies have indicated that bowel wall edema in patients with fluid overload might contribute to bacterial endotoxin translocation [35].

Previous studies about associations between fluid overload and renal outcome do not appear to have sufficiently considered how salt intake influences the effects of excess volume on renal outcome. In our study, no significant difference in renal outcome was seen among Groups 1 and 2. Such findings suggest that fluid overload is unrelated to renal outcome in the absence of high DSI. Although ECW/TBW could also be increased in lean, elderly patients with low ICW [36], patients in Groups 2 and 4 with ECW/TBW above the median were younger and showed relatively higher BMI compared with patients in Groups 1 and 3 with ECW/TBW at or below the median and were thus unlikely to be considered frail. Based on our results that the higher DSI group (>6 g/day) had a higher risk of outcomes than the lower-intake group (≤6 g/day) despite no significant difference according to ECW/TBW, and that patients in Group 3 had the highest risk of all groups, DSI could be considered to be associated with renal outcome independent of volume status.

Differences in outcomes among Groups 3 and 4 despite similar DSI >6 g/day represent an interesting issue. Indeed, differences in DSI were seen between Groups 3 and 4, with median values of 7.76 and 9.15 g/day, respectively. However, DSI as estimated from 24-h urinary sodium excretion may be overestimated, on the basis of the much greater diuretic use in Group 4 than in Group 3. In brief, patients in Group 4 may not have experienced poor renal outcomes because DSI may have been less than estimated. Considering the effect of diuretics use, we performed subgroup analysis in patients without diuretics. Although there is a limit to the interpretation of the results in this small-sample size analysis of 135 patients, median DSI of Group 4 became lower (9.2 to 8.2 g/day), and the point estimate of HR became higher (1.14 to 1.26). It was suggested that patients of falsely high DSI with diuretics use would have a decreased HR in Group 4. Moreover, low dietary adherence or low salt sensitivity might be contributed to the results of a significant association with renal outcome in Group 3, although we could not investigate them in the present study. In addition, recent experimental studies suggest that sodium balance is regulated by other extra-renal mechanisms, and the skin could work as a reservoir of sodium, independent of renal control [37]. Therefore, despite high salt intake, some people may accumulate sodium in the skin without increased volume status. Patients in Group 3 could be included in this population. In several studies using ^23^Na-magnetic resonance imaging, sodium storage in the skin was detected, and skin sodium content was strongly associated with left ventricular mass independent of BP or volume status, even though the detailed mechanisms remain unclear [38]. Although the relationship between sodium storage in the skin and renal outcome is still largely unknown, our results of a significant association with renal outcomes in Group 3 might be associated with stored sodium in the skin. However, this remains purely speculative for now. Further accumulation of results is needed to clarify the associations between sodium storage in the skin and renal outcomes.

## 5. Limitations

Some limitations of this study must be considered. The most serious limitations were the assessment of DSI by a single 24-h urine collection and the short follow-up period. Some reports demonstrated that single 24-h urine collection was insufficient to estimate individual-level long-term DSI [39]. With grouping of exposure based on multiple 24-h urine collections, we might classify patients more correctly with less misclassification than using single collection. Moreover, DSI estimation by single 24-h urine collection at admission could not reflect the estimation during the observation period even if it was evaluated accurately. DSI would be changeable particularly in those with educational hospitalization. However, it was thought noteworthy that a significant difference remained even though all participants were given dietary education in our study. Further studies investigating with multiple urine collections and a longer follow-up period are needed. In addition, urinary sodium excretion might be affected by other causes such as diuretic use and decreased eGFR. A space flight simulation study has also reported that in healthy subjects under controlled sodium intake, urinary sodium excretion changes periodically [40]. Investigating the association between a “correct” salt intake and renal outcomes is considered difficult unless conducted as an interventional study. New tools to evaluate salt intake are expected.

Additionally, the present study retrospectively examined a small cohort from a single institute. Further, it was also thought as a limitation that this population might have had relatively high health literacy, as they voluntarily sought educational admission for CKD. In fact, median DSI was 5.88 g/day, under the widely recommended amount of 6 g/day. However, interestingly, differences in renal outcomes according to DSI remained even in this potentially health-literate population. Regarding the lack of association between ECW/TBW and renal outcome, the higher fluid volume in this study might not have been enough to affect renal outcomes because these patients were not admitted for critical events. Further investigation in patients with a wider range of volume statuses is needed.

Finally, whether ECW/TBW offers an appropriate marker of volume status remains contentious. This value could be affected by age and muscle mass. However, no one parameter can precisely evaluate volume status. We used ECW/TBW as a noninvasive, reproducible, and simple marker of volume status. Further investigation is needed to identify more accurate, easier tools for evaluating volume status.

Regardless of these limitations, it seems very meaningful that the results suggest that patients with appropriate salt intake were associated with favorable renal outcomes despite high volume status, using 24-h urine data collected from 197 patients. Further studies with multiple urine collections and a longer follow-up period in larger populations are expected in the future.

## 6. Conclusions

High salt intake might be associated with poor renal outcomes independent of BP, proteinuria, and volume status.

## Figures and Tables

**Figure 1 nutrients-13-00650-f001:**
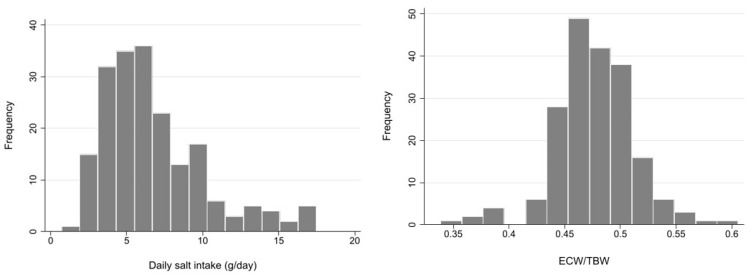
Histogram of daily salt intake and extracellular water (ECW)/total body water (TBW). ECW: extracellular water, TBW: total body water.

**Figure 2 nutrients-13-00650-f002:**
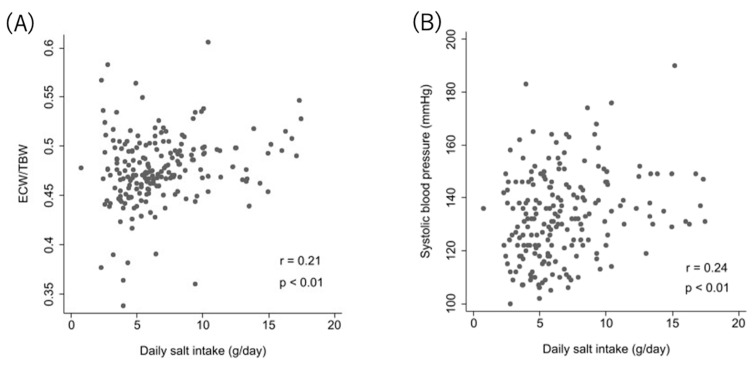
Correlation between DSI and ECW/TBW and systolic blood pressure. Relationship between relative DSI and ECW/TBW (**A**), and between DSI and systolic blood pressure (**B**).

**Figure 3 nutrients-13-00650-f003:**
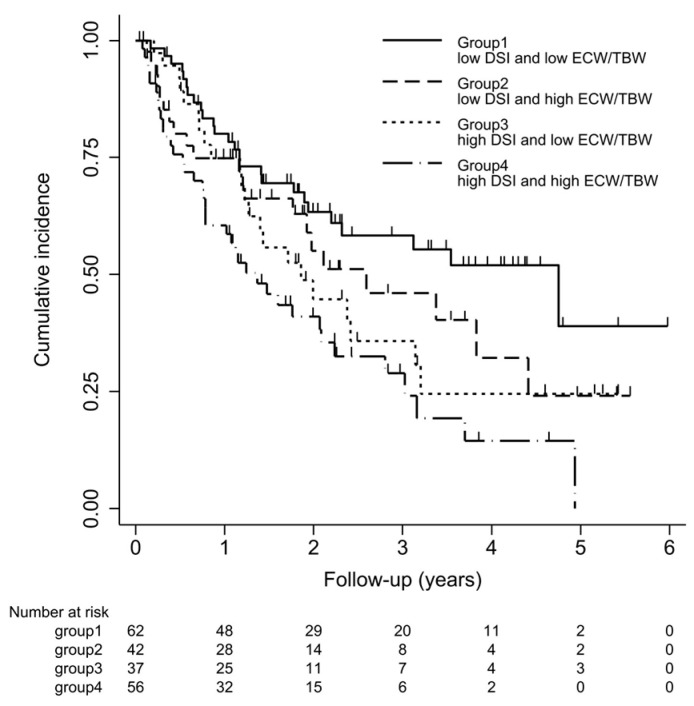
Kaplan–Meier curve for ≥30% decline in eGFR, end-stage renal disease (ESRD), or death. Kaplan–Meier survival curve of outcomes for four groups. Patients were classified into 4 groups according to DSI (≤6 g/day or >6 g/day) and ECW/TBW (≤median or >median). Cumulative incidences (95% confidence interval) of outcomes defined as a ≥30% decline in eGFR from baseline on admission, occurrence of ESRD, or death are shown.

**Figure 4 nutrients-13-00650-f004:**
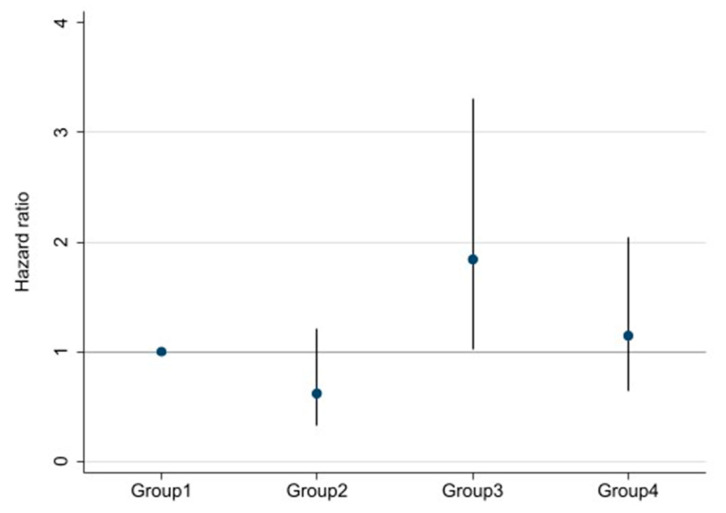
Multivariable-adjusted hazard ratios (HRs) and 95% confidence intervals for ≥30% decline in eGFR, ESRD, or death among four groups divided by both DSI and ECW/TBW. Adjusted for age, sex, estimated glomerular filtration rate, hemoglobin, serum albumin, urine protein, systolic blood pressure, presence/absence of diabetes mellitus, and chronic kidney disease.

**Table 1 nutrients-13-00650-t001:** Baseline characteristics.

Variables	Total *n* = 197		DSI								*p*-Value
			Low ≦ 6.0 g/day				High ≧ 6.0 g/day				
			ECW/TBW								
			Low ≦ 0.48 *n* = 62		High > 0.48 *n* = 42		Low ≦ 0.48 *n* = 37		High > 0.48 *n* = 56		
Age (years)	70.5	(12.1)	73.2	(13.6)	72	(8.6)	69.8	(11.5)	67	(12.3)	0.032 ^b^
Sex, male (*n*(%))	149	(75.6)	43	(69.4)	30	(71.4)	31	(83.8)	45	(80.4)	0.30 ^d^
Body mass inde (kg/m^2^)	24.6	(4.8)	22.1	(3.5)	26	(4.7)	23.7	(3.3)	26.9	(5.4)	<0.001 ^b^
Cause of CKD											
Diabetes mellitus (*n*(%))	62	(31.5)	10	(16.1)	22	(52.4)	8	(21.6)	22	(39.3)	<0.001 ^d^
Nephrosclerosis (*n*(%))	55	(27.9)	20	(32.3)	8	(19.1)	12	(32.4)	15	(26.8)	0.45 ^d^
Chronic glomerulonephritis (*n*(%))	24	(12.2)	11	(17.7)	4	(9.5)	4	(10.8)	5	(8.9)	0.49 ^d^
Others (*n*(%))	56	(28.4)	21	(33.9)	8	(19.1)	13	(35.1)	14	(25.0)	0.28 ^d^
Comorbidities											
Hypertension (*n*(%))	186	(94.4)	55	(88.7)	39	(92.9)	37	(100)	55	(98.2)	0.005 ^d^
Diabetes mellitus (*n*(%))	92	(46.7)	18	(29.0)	27	(64.3)	12	(32.4)	35	(62.5)	<0.001 ^e^
Cerebrovascular disease (*n*(%))	65	(33.0)	24	(38.7)	9	(21.4)	16	(43.2)	16	(28.6)	0.13 ^e^
ECW/TBW	0.48	(0.04)	0.45	(0.03)	0.5	(0.03)	0.46	(0.02)	0.5	(0.02)	<0.001^b^
Laboratory test											
Hemoglobin (g/dL)	11.4	(1.8)	11.4	(1.6)	11.2	(1.8)	11.9	(1.8)	11.2	(1.8)	0.27 ^b^
Serum albumin (mg/dL)	3.9	(0.5)	4	(0.4)	3.8	(0.5)	4	(0.4)	3.7	(0.5)	<0.001 ^b^
eGFR (mL/min/1.73m^2^)	24.2	(11.1)	26	(10.7)	23.1	(10.9)	24.9	(11.7)	22.4	(11)	0.29 ^b^
45–59 mL/min/1.73m^2^ (*n*(%))	8	(4.1)	3	(4.8)	2	(4.8)	1	(2.7)	2	(3.6)	0.80 ^d^
30–44 mL/min/1.73m^2^ (*n*(%))	44	(22.3)	15	(4.2)	9	(21.4)	10	(27.0)	10	(17.9)	
15–29 mL/min/1.73m^2^ (*n*(%))	100	(50.8)	35	(56.5)	19	(45.2)	17	(46.0)	29	(51.8)	
<15 mL/min/1.73m^2^ (*n*(%))	45	(22.8)	9	(14.5)	12	(28.6)	9	(24.3)	15	(26.8)	
Urine protein (g/day ^a^) [IQR]	0.9	[0.2, 2.2]	0.5	[0.1, 1.0]	0.8	[0.3, 2.3]	0.7	[0.2, 1.8]	2.2	[0.9, 4.0]	<0.001 ^c^
DSI (g/day ^a^) [IQR]	5.9	[4.4, 8.2]	4.5	[3.7, 5.4]	4.2	[3.2, 5.1]	7.8	[6.7, 9.9]	9.2	[7.5, 11.2]	<0.001 ^c^
Renin-angiotensin inhibitor (*n*(%))	133	(67.5)	39	(62.9)	31	(73.8)	23	(62.2)	40	(71.4)	0.53 ^d^
Diuretics (*n*(%))	62	(31.5)	14	(22.6)	15	(35.7)	5	(13.5)	28	(50.0)	0.001 ^e^
Systolic blood pressure (mmHg)	133	(16)	128	(16)	132	(16)	132	(13)	140	(17)	<0.001 ^b^
Diastolic blood pressure (mmHg)	74.3	(8.4)	72.6	(8.2)	72.6	(8.3)	76.1	(7.7)	76.4	(8.8)	0.02 ^b^

Values are presented as mean (standard deviation) or number (%), unless otherwise specified. ^a^ Median (interquartile range); ^b^ ANOVA; ^c^ Kruskal–Wallis test; ^d^ Fisher’s exact test; ^e^ chi-square test. IQR: interquartile range; CKD: chronic kidney disease; ECW: extracellular water; TBW: total body water; eGFR: estimated glomerular filtration rate; DSI: daily salt intake; body mass index is calculated as the weight in kilograms divided by the square of the height in meters. eGFR (mL/min/1.73 m^2^) = 194 × serum creatinine (−1.094) × age (−0.287) × 0.739 (if female). DSI is estimated by measuring 24-h urinary sodium excretion.

**Table 2 nutrients-13-00650-t002:** Incidence of eGFR ≧30 % decline or renal replacement therapy or death.

	*n*	Observed Time (Years)		Incidence				Incident Rate (/100 Person-Year)	[95% CI]
		Median	IQR	Total *n* (%)	≧30% eGFR Decline *n* (%)	ESRD *n* (%)	Death *n* (%)		
Total	197	1.4	[0.7, 2.4]	107(54.3)	98(49.7)	6(3.0)	3(1.5)	29.8	[24.6, 38.0]
Group1	62	1.9	[1.1, 3.5]	26(41.9)	23(37.1)	2(3.2)	1(1.6)	18.7	[12.7, 27.4]
Group2	42	1.5	[0.4, 2.3]	21(50.0)	21(50.0)	0	0	27.9	[18.2, 42.8]
Group3	37	1.4	[0.8, 2.4]	22(59.5)	18(48.7)	2(5.4)	2(5.4)	32.5	[21.4, 49.4]
Group4	56	1.1	[0.4, 2.1]	38(67.9)	36(64.3)	2(3.6)	0	49.2	[35.8, 67.6]

IQR: interquartile range; 95% CI: 95% confidence interval.

## Data Availability

The data presented in this study are available on request from the corresponding author.

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
