# Peer review of "Association between Daily Urinary Sodium Excretion, Ratio of Extracellular Water-to-Total Body Water Ratio, and Kidney Outcome in Patients with Chronic Kidney Disease"

_nutrients, 2021, doi:10.3390/nu13020650_

Round 1
Reviewer 1 Report
Kohatsu et al. have investigated the association of volume status and dietary salt intake (DSI) with (renal) outcomes in chronic kidney disease (CKD) patients. The authors used a retrospective cohort study and patients were derived from medical records at the st. Marianna University School of Medicine Hospital. They included 197 patients with stage 3-5 CKD that were hospitalized between January 2011 and April 2019 to receive health education on their condition.
DSI was estimated at hospital admission using the urinary sodium excretion of a single 24-h urine collection. Volume status was assessed with the ratio of extracellular water to total body water (ECW/TBW) as measured by bioelectrical impedance analysis. Outcomes were defined as a ³30% decline in eGFR from baseline, occurrence of end-stage renal or death.
Patients were divided up in to low (≤6 g/d) vs high DSI (>6 g/d) groups and low (≤0.48) vs high ECW/TBW (>0.48) groups. These were further split up in to subgroups: low DSI and low ECW/TBW (group 1), low DSI and high ECW/TBW (group 2), high DSI and low ECW/TBW (group 3), high DSI and high ECW/TBW (group 4). The authors found that high DSI was associated with worse outcomes (HR 1.69, 95% CI 1.12-2.57; p=0.01) than low DSI. No significant association was found between renal outcomes and ECW/TBW. When analyzing the subgroups using a Cox proportional hazards analysis, they found that group 3 (high DSI and low ECW/TBW) had an 1.84-fold (95%CI 1.03–3.30-fold; p=0.04) excess risk of renal outcome occurrence.
Major
An important problem of this study is the measurement of one the primary variables. The authors used a single 24-h urine collection for estimation of individual-level long-term DSI. A large body of evidence over the past few years has demonstrated that this method is inaccurate and significantly affects the relation between sodium excretion and the investigated outcome. Using multiple collections instead of a single collection significantly alters the observed association (Circulation. 2018 Apr 3;137(14):1538-1539). This has led to an international consensus that a single 24-h urine collection should not be used for assessment of sodium intake in relation to long-term health outcomes (J Clin Hypertens (Greenwich). 2019 Jun;21(6):700-709). If additional 24-hour urine collection data is available it would be strongly recommended to include this.
The median follow-up in this study was 1.4 years (IQR 0.7–2.4). During this time 54.3% of the included patients reached one of the outcomes of interest. Even if DSI estimation by single 24-h urine collection at admission was accurate, it only reflects DSI at that given moment. Would it be possible that patients changed their diet before hospitalization due to social desirability? Did they change their diet after hospitalization as a result of the education they received? All of this is not reflected in the single 24-h urine that was collected upon admission. Further, it seems unlikely that DSI can have an substantial effect on (renal) outcomes over such a short time span. The short follow-up should be mentioned as a limitation, but it would be better if authors can include urinary sodium excretion data from before (and after) admission into their analysis.
The use of an inaccurate sodium intake estimation method may also explain the counterintuitive difference in outcomes between groups 3 and 4. Especially, if you consider that DSI is significantly higher in group 4. The authors suggest urinary sodium excretion may be overestimated in group 4, given the much greater use of diuretics. Maybe they can consider correcting for diuretic use in their model?
Minor
28: “[…] various adverse health outcomes, such as stomach cancer [1], osteoporosis [2], and kidney stones [3]. Of these, the most crucial issues are undoubtedly new onset or worsening of hypertension […]”
It is suggested to rephrase this sentence as it is awkwardly written. Authors comprise a list of several health outcomes and refer back to this list using “of these” but then state hypertension and CVD, which were not mentioned before.
42: Use of “also” in first sentence of paragraph reads awkward. It is suggested that this sentence is rephrased.
46 – 49: It is suggested that these sentences are slightly rephrased for reader convenience.
53: please clarify what guideline is being used for CKD staging.
60: “[…] to provide patients with the opportunity to use the official departmental website […]”
In this sentence “with” is redundant.
68: “[…] or for whom eGFR could not be followed after discharge were excluded.”
Suggestion to rephrase this as: “or for whom eGFR was not followed up after discharge were excluded.”
69: “Among these 204 patients, 3 patients who met the definition of oliguria and 4 patients for home eGFR could not be followed for >1 day were excluded.”
It is suggested to rephrase this sentence as it reads awkward.
74: a comma should be placed between “body mass index (BMI)” and “etiology of CKD”.
86: please include a statement on the amount of 24-h urine collections used for estimation of dietary salt intake.
117: “Differences between the four groups shown in 2.3.2. c) […]”
Suggestion to use the term groups (high vs low DSI and high vs low ECW/TBW) and subgroups (combined groups) for reader convenience.
125: Please clarify the meaning of the abbreviation “UP” as it has not been stated yet.
140: “[…] significantly lower BMI and UP. than […]”
Incorrect placement of period between “UP” and “than”.
140: “Furthermore, diabetic nephropathy was less frequent and nephrosclerosis and chronic glomerulonephritis were more frequent as etiologies of CKD, and patients with hypertension and high SBP were less frequent in Group 1. Meanwhile, patients in Group 4 were significantly younger, had higher BMI and frequency of DM, and significantly higher UP and daily salt intake than the other groups.”
It is suggested to rephrase these sentences in order to more clearly state the differences between different groups for the reader.
235: please state reference
246: dash in “bowel-wall edema” is redundant
279: Suggestion to replace “Actually” by “In fact”
Author Response
We apologize for taking so long to reply. Thank you for giving us the opportunity to strengthen our manuscript with your valuable comments and queries. We have worked hard to incorporate your feedback and hope that these revisions persuade you to accept our submission.
An important problem of this study is the measurement of one the primary variables. The authors used a single 24-h urine collection for estimation of individual-level long-term DSI. A large body of evidence over the past few years has demonstrated that this method is inaccurate and significantly affects the relation between sodium excretion and the investigated outcome. Using multiple collections instead of a single collection significantly alters the observed association (Circulation. 2018 Apr 3;137(14):1538-1539). This has led to an international consensus that a single 24-h urine collection should not be used for assessment of sodium intake in relation to long-term health outcomes (J Clin Hypertens (Greenwich). 2019 Jun;21(6):700-709). If additional 24-hour urine collection data is available it would be strongly recommended to include this.
Thank you for your suggestion. You have raised very important points. In this study, we used only single 24-h urine collection measured at admission. In fact, however, we measured 24-h urine collection twice, at the 2nd and 4th day during educational hospitalization for CKD in our hospital. However, we used the data of only the 2nd day, not the 4th day because the data of the 4th day was thought to be affected by diet during hospitalization (salt intake was restricted under 6 g/day during hospitalization) and inappropriate to estimate. Indeed, DSI estimated by 24-h urine salt excretion at the 4th day was 4.29[3.2, 5.8]g/day, significantly less than that of the 2nd day, 5.9[4.4, 8.2]g/day. In addition, with regard to measurement of 24-h salt excretion during hospitalization, it was also the fact that 24-h urine could be collected more correctly than collection at outpatient visit even for just a single measurement. As you indicated, grouping of exposure based on multiple urine collections could classify patients more correctly and prevent misclassification than using single collection. We described that points as limitation in this article.
Indeed, it was suggested that single measurement of urine sodium excretion was not generally sufficient to evaluate the estimation of long-term individual’s usual salt intake. However, it was also suggested that this is not the case for the estimation of the average sodium intake of a study population. Moreover, it was indicated that single measurement of urine sodium excretion was sufficient to detect the direction of the relation between sodium intake and health outcomes in observational studies in “letter to the editor” of Circulation 2017. Therefore, we considered that it has been controversial how sodium intake with respect to long-term outcomes can most reliably be assessed.
The median follow-up in this study was 1.4 years (IQR 0.7–2.4). During this time 54.3% of the included patients reached one of the outcomes of interest. Even if DSI estimation by single 24-h urine collection at admission was accurate, it only reflects DSI at that given moment. Would it be possible that patients changed their diet before hospitalization due to social desirability? Did they change their diet after hospitalization as a result of the education they received? All of this is not reflected in the single 24-h urine that was collected upon admission. Further, it seems unlikely that DSI can have an substantial effect on (renal) outcomes over such a short time span. The short follow-up should be mentioned as a limitation, but it would be better if authors can include urinary sodium excretion data from before (and after) admission into their analysis.
Thank you for your valuable advice. As you indicated, DSI estimation by single 24-h urine collection at admission could not reflect the estimation during observation period even if it was evaluated accurately. It was considered limiting point, the characteristic of observational study. In particular, given the population with educational hospitalization, DSI would be changeable after hospitalization, which was assumed to make difference between groups smaller. However, it was thought noteworthy that significant difference remained even though all participants were given dietary education.
We described that as limitation.
The use of an inaccurate sodium intake estimation method may also explain the counterintuitive difference in outcomes between groups 3 and 4. In particular, if you consider that DSI is significantly higher in group 4. The authors suggest urinary sodium excretion may be overestimated in group 4, given the much greater use of diuretics. Maybe they can consider correcting for diuretic use in their model?
Thank you for providing these insights. Considering the effect of diuretics use as you mentioned, we performed subgroup analysis in patients without diuretics. Although there is limit to interpretation of the results because of small sample size analysis of 135 patients, median DSI of Group 3 became lower (9.2 to 8.2), and the point estimate of HR became higher (1.14 to 1.26). It was thought that patients of false high DSI with diuretics use would decrease HR. We added the results of subgroup analysis in this article.
Minor
28: “[…] various adverse health outcomes, such as stomach cancer [1], osteoporosis [2], and kidney stones [3]. Of these, the most crucial issues are undoubtedly new onset or worsening of hypertension […]”
It is suggested to rephrase this sentence as it is awkwardly written. Authors comprise a list of several health outcomes and refer back to this list using “of these” but then state hypertension and CVD, which were not mentioned before. We agree with you and have incorporated this suggestion throughout our paper.
42: Use of “also” in first sentence of paragraph reads awkward. It is suggested that this sentence is rephrased.
We agree with your assessment, and rewritten this sentence.
46 – 49: It is suggested that these sentences are slightly rephrased for reader convenience.
We agreed your advices. We have rewritten these sentences easier to understand for reader.
53: please clarify what guideline is being used for CKD staging.
We were very sorry to fail to mention it. We added reference about CKD staging.
60: “[…] to provide patients with the opportunity to use the official departmental website […]”
In this sentence “with” is redundant.
Agreed. We have delated the word “with” in this sentence.
68: “[…] or for whom eGFR could not be followed after discharge were excluded.”
Suggestion to rephrase this as: “or for whom eGFR was not followed up after discharge were excluded.”
We agree with your suggestion and have rephrased the sentence as you indicated.
69: “Among these 204 patients, 3 patients who met the definition of oliguria and 4 patients for home eGFR could not be followed for >1 day were excluded.”
It is suggested to rephrase this sentence as it reads awkward.
Thank you for your suggestion. We have changed this sentence, “followed for > 1day” to “followed after discharge”.
74: a comma should be placed between “body mass index (BMI)” and “etiology of CKD”.
Agreed. We have placed comma between them.
86: please include a statement on the amount of 24-h urine collections used for estimation of dietary salt intake.
Agreed. We have included a statement there.
117: “Differences between the four groups shown in 2.3.2. c) […]”
Suggestion to use the term groups (high vs low DSI and high vs low ECW/TBW) and subgroups (combined groups) for reader convenience.
Thank you for your suggestion. We have rewritten the description of each group to be more in line with your comments, “high vs low DSI and high vs low ECW/TBW”. Description of 4 Groups in Figure 3 was also changed.
125: Please clarify the meaning of the abbreviation “UP” as it has not been stated yet.
We are sorry to fail to describe it. We have clarified the meaning of that.
140: “[…] significantly lower BMI and UP. than […]”
Incorrect placement of period between “UP” and “than”.
We agree with you and have removed the period.
140: “Furthermore, diabetic nephropathy was less frequent and nephrosclerosis and chronic glomerulonephritis were more frequent as etiologies of CKD, and patients with hypertension and high SBP were less frequent in Group 1. Meanwhile, patients in Group 4 were significantly younger, had higher BMI and frequency of DM, and significantly higher UP and daily salt intake than the other groups.”
It is suggested to rephrase these sentences in order to more clearly state the differences between different groups for the reader.
Thank you for your suggestion. We have rewritten these sentences to describe more clearly the differences between different groups.
235: please state reference
We are sorry to fail to state. We have added the reference.
246: dash in “bowel-wall edema” is redundant
Thank you for your suggestion. We have removed the dash.
279: Suggestion to replace “Actually” by “In fact”
Thank you for your suggestion. We have replaced the term “Actually” with “In fact”.
Finally, we have added a sentence “This research was supported by AMED under Grant Numbers JP20ek0310010h0003.”in acknowledgments. We apologize that we failed to write in the manuscript previously submitted.
In addition, we used “DSI” as abbreviation of “daily salt intake”. Because some parts remained without abbreviation in submitted manuscript, then we have modified them in this article.
Reviewer 2 Report
I must apologise for the limited comments, but time-pressures precluded a more thorough assessment. I would say overall that I found this article to be very interesting. The only reference to nutrition is the estimated dietary salt intake. Unfortunately dietary salt was not available (I presume) to the authors and so the intake estimate is based upon 24hr excretion. Whilst under some circumstances, this may be a valid approach, it is less valuable in people with declining renal function which this group clearly represent (eGFR mean of 24.2 ml/min/1.73m2). So the focus of the article seems more based on renal sodium excretion capacity, which in itself would seem a perfectly appropriate central point making this paper far more appropriate (in my opinion) to a renal or physiology journal. The results presented, regardless of this point are very well indicated and show some interesting, expected outcomes. I have included a couple of specific points below that I think warrant attention.
Line 85-89 – Estimation of salt intake from urinary sodium excretion. Did the authors validate this mode of estimation of salt intake in this or previous studies? If not, it is necessary to provide citations for studies that indicate the validity of this measurement. Declining renal function is very likely to impact the renal ability to excrete sodium. Therefore, the measurement of 24hr excretion may be of limited value in determining intake. It might be better to simply focus the paper around 24 hr excretion. Increasing intake is likely to increase excretion, but in a CKD population, this is not necessarily going to be a linear relationship.
Results
Table 2 – this would be more informative if it included information about each separate outcome rather than all combined.
Author Response
We apologize for taking so long to reply. Thank you for giving us the opportunity to strengthen our manuscript with your valuable comments and queries. We have worked hard to incorporate your feedback and hope that these revisions persuade you to accept our submission.
I must apologise for the limited comments, but time-pressures precluded a more thorough assessment. I would say overall that I found this article to be very interesting. The only reference to nutrition is the estimated dietary salt intake. Unfortunately dietary salt was not available (I presume) to the authors and so the intake estimate is based upon 24hr excretion. Whilst under some circumstances, this may be a valid approach, it is less valuable in people with declining renal function which this group clearly represent (eGFR mean of 24.2 ml/min/1.73m2). So the focus of the article seems more based on renal sodium excretion capacity, which in itself would seem a perfectly appropriate central point making this paper far more appropriate (in my opinion) to a renal or physiology journal. The results presented, regardless of this point are very well indicated and show some interesting, expected outcomes. I have included a couple of specific points below that I think warrant attention.
Thank you for your suggestion. With the aim of validating the effect of salt restriction in patients with CKD stage3-5 on their renal outcome, we have submitted to this journal. We have clarified that there was limitation to evaluate salt intake in discussion.
Line 85-89 – Estimation of salt intake from urinary sodium excretion. Did the authors validate this mode of estimation of salt intake in this or previous studies? If not, it is necessary to provide citations for studies that indicate the validity of this measurement. Declining renal function is very likely to impact the renal ability to excrete sodium. Therefore, the measurement of 24hr excretion may be of limited value in determining intake. It might be better to simply focus the paper around 24 hr excretion. Increasing intake is likely to increase excretion, but in a CKD population, this is not necessarily going to be a linear relationship.
Thank you for your suggestion. Indeed, it has been probably unknown whether 24-h urine sodium excretion is sufficient to evaluate DSI correctly in CKD patients, however, there were several reports which investigated the effect of urinary sodium excretion on ESKD or cardiovascular outcomes in patients with CKD (reference from[7][10]in article). As you indicated, DSI estimation by 24-h urine sodium excretion could be influenced by declining renal function, and described that as limitation.
Results
Table 2 – this would be more informative if it included information about each separate outcome rather than all combined.
We agree with your suggestion and have added each separate outcome in Table 2.
Finally, we have added a sentence “This research was supported by AMED under Grant Numbers JP20ek0310010h0003.”in Acknowledgments. We apologize that we failed to write in the manuscript previously submitted. In addition, we used “DSI” as abbreviation of “daily salt intake”. Because some parts remained without abbreviation in submitted manuscript, then we have modified them in this article.
Round 2
Reviewer 1 Report
Major points:
Firstly, compliments to the authors for explaining their points, carefully editing their manuscript and including additional analyses.
Secondly, it is correct that a single 24-hour urine collection is sufficient for estimating average sodium intake in a study population. This means that one can use this method if they were merely interested in what the average sodium intake of CKD patients before patient education is. The authors could, hypothetically, repeat the measurement three months after admission to evaluate average sodium intake of CKD patients after patient education. However, they are investigating the relationship between DSI and ECW/TBW and health/renal outcomes. Health outcomes, even in a study population, are inseparably connected to the individual. This is demonstrated by survival analyses (like the Kaplan-Meier and Cox regression), which use individual measurements and individual outcomes to produce a curve or hazard ratio. However, we know that the measurements used in this study are not sufficient for estimation of individual-level sodium intake. Maybe the authors can approach the data in a manner that emphasizes less on individual sodium intake and health outcomes as their method of determining individual-level sodium intake is insufficient?
Thirdly, if understood correctly, the authors are referring to the “letter to the editor” of Circulation 2017 by Graudal and Mente. This letter argues that a single 24-hour urine sodium excretion estimate is sufficient to detect the direction of the relation between sodium intake and health outcomes in observational studies. In a “response to letter to the editor” of Circulation 2018 by Olde Engberink et al, this point is extensively rebutted. Further, The International Consortium for Quality Research on Dietary Sodium/Salt (TRUE) position statement on the use of 24‐hour, spot, and short duration (<24 hours) timed urine collections to assess dietary sodium intake, which was published in the Journal of Clinical Hypertension 2019, states that for estimation of long-term sodium intake, which is of interest when examining chronic disease outcomes, 24-hour urine collections need to be repeated throughout the study timeframe. However, given the relatively short follow-up and limited 24-h urine collections it is understandable that this is not feasible in this study. Again, by approaching the data in a manner that emphasizes less on these long-term individual health outcomes the authors may have an interesting manuscript at hands, but currently the scientific soundness is still lacking.
Finally, despite the limitations caused by short follow-up and the use of a single 24-h urine collection, some of the data is noteworthy. As mentioned by the authors, it is interesting that even though all patients received the same patient education hazard risks were significantly different. Do the authors think this can be explained by e.g. difference in ECW/TBW, low dietary adherence, low salt sensitivity or maybe something else? It could be interesting to explore this further to generate a more robust and comprehensive analysis of the data.
Minor points:
47-49: “In those studies, however, how salt intake and fluid overload interacted in their association with renal outcome each other was not evaluated sufficiently”
If understood correctly what the authors are trying to say, “renal outcome” should be plural and “each other” needs to be removed in this sentence.
147-150: “Furthermore, patients in Group 1 were less frequent diabetic nephropathy, whereas more frequent nephrosclerosis and chronic glomerulonephritis as etiologies of CKD, and less number of people had hypertension in Group 1.”
The following edit for this sentence is given for consideration: “Furthermore, etiologies of CKD in Group 1 were more frequently nephrosclerosis and chronic glomerulonephritis and less frequently diabetic nephropathy. In addition, a smaller number of people had hypertension in Group 1.”
211-214: in these sentences “salt intake” has not yet been replaced by “DSI”.
306-310: “Considering the effect of diuretics use, we performed subgroup analysis in patients without diuretics. Although there is limit to interpretation of the results in small sample size analysis of 135 patients, median DSI of Group 3 became lower, and the point estimate of HR became higher. It was suggested that patients of false high DSI with diuretics use would decrease HR in Group 3.”
The authors mention that interpretation of the results of the subgroup analysis in patients without diuretics was limited due to the small sample size. Does this mean that the decrease in median DSI and increase in point estimate of HR that were observed in group 3 were not significant? Furthermore, how did exclusion of patients without diuretics affect the associations in other groups and how did it affect overall association with outcomes? What seems confusing is that in the discussion the authors argue that group 4 may have false high DSI as half of the patients in this group use diuretics. Please clarify this inconsistency.
322 “ […] using 24-h urine data (as the gold-standard method to estimate salt intake) […]”
This sentence may be perceived as slightly misleading, as a single 24-h urine collection is only suitable to estimate average population intake. The fact that 24-h urine collections are most often used in clinical studies does not necessarily mean that this is the gold-standard method. In my opinion, a gold-standard method for DSI would be represented by 7-9 24-h collections.
Author Response
Thank you for giving us the opportunity to resubmit our article for further consideration.
We have incorporated changes that reflect the detailed suggestions you have graciously provided. We also look forward to hearing from you regarding our submission. We would be glad to respond to any further questions and comments that you may have.
Major points:
Firstly, compliments to the authors for explaining their points, carefully editing their manuscript and including additional analyses.
Secondly, it is correct that a single 24-hour urine collection is sufficient for estimating average sodium intake in a study population. This means that one can use this method if they were merely interested in what the average sodium intake of CKD patients before patient education is. The authors could, hypothetically, repeat the measurement three months after admission to evaluate average sodium intake of CKD patients after patient education. However, they are investigating the relationship between DSI and ECW/TBW and health/renal outcomes. Health outcomes, even in a study population, are inseparably connected to the individual. This is demonstrated by survival analyses (like the Kaplan-Meier and Cox regression), which use individual measurements and individual outcomes to produce a curve or hazard ratio. However, we know that the measurements used in this study are not sufficient for estimation of individual-level sodium intake. Maybe the authors can approach the data in a manner that emphasizes less on individual sodium intake and health outcomes as their method of determining individual-level sodium intake is insufficient?
Thank you for your suggestion. As you mentioned, our study was analyzed using individual measurements and outcomes, and investigated individual’s health outcomes, therefore, we should have considered the estimation of individual-level sodium intake. Then, sufficiency for estimating average sodium intake in a study population could not be a reason to use a single 24-hour urine collection in our study. We apologize that we could not have understood correctly and send an inappropriate answer. In our study, we have used only single 24h urine collection and described as limitation.
Thirdly, if understood correctly, the authors are referring to the “letter to the editor” of Circulation 2017 by Graudal and Mente. This letter argues that a single 24-hour urine sodium excretion estimate is sufficient to detect the direction of the relation between sodium intake and health outcomes in observational studies. In a “response to letter to the editor” of Circulation 2018 by Olde Engberink et al, this point is extensively rebutted. Further, The International Consortium for Quality Research on Dietary Sodium/Salt (TRUE) position statement on the use of 24‐hour, spot, and short duration (<24 hours) timed urine collections to assess dietary sodium intake, which was published in the Journal of Clinical Hypertension 2019, states that for estimation of long-term sodium intake, which is of interest when examining chronic disease outcomes, 24-hour urine collections need to be repeated throughout the study timeframe. However, given the relatively short follow-up and limited 24-h urine collections it is understandable that this is not feasible in this study. Again, by approaching the data in a manner that emphasizes less on these long-term individual health outcomes the authors may have an interesting manuscript at hands, but currently the scientific soundness is still lacking.
We could understand that single 24-hour urine collection was insufficient to estimate individual-level sodium intake, and should be repeated. As you indicated, short follow-up and only single 24-h urine collection were limiting points which could not be ignored in our study. We described them as limitation in article.
In addition, considering these limiting points, we have changed our conclusion to weaker argument, rephrasing from “appears to be” to “might be”.
Finally, despite the limitations caused by short follow-up and the use of a single 24-h urine collection, some of the data is noteworthy. As mentioned by the authors, it is interesting that even though all patients received the same patient education hazard risks were significantly different. Do the authors think this can be explained by e.g. difference in ECW/TBW, low dietary adherence, low salt sensitivity or maybe something else? It could be interesting to explore this further to generate a more robust and comprehensive analysis of the data.
Thank you for your providing these insights. As you indicated, low dietary adherence or low salt sensitivity could be contributed to our results of a significant association with renal outcome in Group 3, although we could not investigate them in our study. With regard to difference in ECW/TBW, we thought it would not be associated because there was no significant difference in renal outcome among ECW/TBW high, low groups in our analysis. Moreover, as we mentioned in article, sodium storage in the skin might be occurred in Group3 and associated with the poor outcome of Group3, although it remains a matter of speculation. We added these considerations as residual confounding factors in discussion.
Minor points:
47-49: “In those studies, however, how salt intake and fluid overload interacted in their association with renal outcome each other was not evaluated sufficiently”
If understood correctly what the authors are trying to say, “renal outcome” should be plural and “each other” needs to be removed in this sentence.
We agreed with your suggestion and have changed “renal outcome” to be plural and removed “each other”.
147-150: “Furthermore, patients in Group 1 were less frequent diabetic nephropathy, whereas more frequent nephrosclerosis and chronic glomerulonephritis as etiologies of CKD, and less number of people had hypertension in Group 1.”
The following edit for this sentence is given for consideration: “Furthermore, etiologies of CKD in Group 1 were more frequently nephrosclerosis and chronic glomerulonephritis and less frequently diabetic nephropathy. In addition, a smaller number of people had hypertension in Group 1.”
Your suggested sentence is easier to understand. We changed the sentence as you suggested.
211-214: in these sentences “salt intake” has not yet been replaced by “DSI”.
Thank you for your indication. Daily salt intake (estimated by single 24-h salt excretion in our study) has been abbreviated as “DSI” in this article, however, we did not apply to the same or similar words used in other reports because we were afraid that what they mean or how they were calculated or estimated would be differed by studies.
306-310: “Considering the effect of diuretics use, we performed subgroup analysis in patients without diuretics. Although there is limit to interpretation of the results in small sample size analysis of 135 patients, median DSI of Group 3 became lower, and the point estimate of HR became higher. It was suggested that patients of false high DSI with diuretics use would decrease HR in Group 3.”
The authors mention that interpretation of the results of the subgroup analysis in patients without diuretics was limited due to the small sample size. Does this mean that the decrease in median DSI and increase in point estimate of HR that were observed in group 3 were not significant? Furthermore, how did exclusion of patients without diuretics affect the associations in other groups and how did it affect overall association with outcomes? What seems confusing is that in the discussion the authors argue that group 4 may have false high DSI as half of the patients in this group use diuretics. Please clarify this inconsistency.
We are very sorry to have written a wrong number of Group. Correctly, it was not Group 3, but Group 4. We would like to write “median DSI of Group 4 became lower (9.2g/day to 8.2g/day), and the point estimate of HR became higher (1.14 to 1.26). It was suggested that patients of falsely high DSI with diuretics use would decrease HR in Group 4.” We apologize for having caused confusion because of our mistake.
322 “ […] using 24-h urine data (as the gold-standard method to estimate salt intake) […]”
This sentence may be perceived as slightly misleading, as a single 24-h urine collection is only suitable to estimate average population intake. The fact that 24-h urine collections are most often used in clinical studies does not necessarily mean that this is the gold-standard method. In my opinion, a gold-standard method for DSI would be represented by 7-9 24-h collections.
We agreed with you. Then, we have removed the sentence within parenthesis. In our study, multiple 24-h urine collection could not be performed and we described that as limitation.